# Current Therapeutics for COVID-19, What We Know about the Molecular Mechanism and Efficacy of Treatments for This Novel Virus

**DOI:** 10.3390/ijms23147702

**Published:** 2022-07-12

**Authors:** Divya Narayanan, Tanyalak Parimon

**Affiliations:** 1General Internal Medicine, Department of Internal Medicine, Cedars Sinai Medical Center, Los Angeles, CA 90048, USA; 2Pulmonary and Critical Care Division, Department of Medicine, Cedars Sinai Medical Center, Los Angeles, CA 90048, USA; tanyalak.parimon@cshs.org

**Keywords:** COVID-19, SARS-CoV-2, therapeutics

## Abstract

Severe acute respiratory syndrome-coronavirus 2 (SARS-CoV-2) has caused significant morbidity and mortality worldwide. Though previous coronaviruses have caused substantial epidemics in recent years, effective therapies remained limited at the start of the Coronavirus disease 19 (COVID-19) pandemic. The emergence and rapid spread throughout the globe of the novel SARS-CoV-2 virus necessitated a rapid development of therapeutics. Given the multitude of therapies that have emerged over the last two years and the evolution of data surrounding the efficacy of these therapies, we aim to provide an update on the major clinical trials that influenced clinical utilization of various COVID-19 therapeutics. This review focuses on currently used therapies in the United States and discusses the molecular mechanisms by which these therapies target the SARS-CoV-2 virus or the COVID-19 disease process. PubMed and EMBASE were used to find trials assessing the efficacy of various COVID-19 therapies. The keywords SARS-CoV-2, COVID-19, and the names of the various therapies included in this review were searched in different combinations to find large-scale randomized controlled trials performed since the onset of the COVID-19 pandemic. Multiple therapeutic options are currently approved for the treatment of SARS-CoV-2 and prevention of severe disease in high-risk individuals in both in the inpatient and outpatient settings. In severe disease, a combination of antiviral and immunomodulatory treatments is currently recommended for treatment. Additionally, anti-viral agents have shown promise in preventing severe disease and hospitalization for those in the outpatient setting. More recently, current therapeutic approaches are directed toward early treatment with monoclonal antibodies directed against the SARS-CoV-2 virus. Despite this, no treatment to date serves as a definitive cure and vaccines against the SARS-CoV-2 virus remain our best defense to prevent further morbidity and mortality.

## 1. Introduction

Since December 2019, Coronavirus disease 2019 (COVID-19) has caused significant morbidity and mortality worldwide, infecting over 500 million people, and causing 6.4 million deaths to date [1]. In the United States alone there have been over 85 million cases and over 1 million deaths associated with COVID-19 [1]. Over the last two years, the incidence of COVID-19 has varied, often coinciding with the emergence of new strains of SARS-CoV-2 [2].

Coronaviruses consist of four structural proteins: the nucleocapsid (N), membrane (M), envelope (E) and spike (S) proteins [3,4]. Prior studies have indicated that the structural spike protein on the virus surface mediates entry of SARS-CoV-2 into human cells via binding to the angiotensin-converting enzyme 2 (ACE2) receptor [4]. Although the ACE2 receptor is found in all organs of the body, it is particularly abundant on alveolar epithelial cells, which enables the lungs to act as the primary entry point for the virus into the human body. Due to the ubiquitous expression of the ACE2 receptor on human cells, the host response to SARS-CoV-2 ranges from minimal respiratory symptoms to severe conditions, such as acute respiratory distress syndrome (ARDS) and multisystem organ dysfunction [5]. Features attributed to the severe phenotype of COVID-19 experienced by some individuals include the severity of immune cell depletion and degree of systemic inflammation produced by the virus. Research suggests that the spike protein is the primary determinant of SARS-CoV-2 diversity [3]. Additionally, the spike protein serves as the primary means by which the host immune system detects the virus. Variations in the spike protein, therefore, lead to both variable transmissibility and infectivity of different viral strains [6]. Many strains have little to no clinical importance. Since the onset of the COVID-19 pandemic, however, there have been four primary variants (alpha, beta, delta and omicron) of COVID-19 that the World Health Organization (WHO) has labeled variants of concern due to their increase in transmissibility, increase in disease severity, or both [7].

Early in the pandemic, few treatment options were available. Existing anti-viral therapies were initially repurposed to treat hospitalized individuals with severe disease. These anti-viral agents, such as RNA polymerase and protease enzymes, functioned by blocking viral genomic replication; however, they showed limited benefit in the severely ill population [8,9]. Early observations demonstrated that the severity of the systemic inflammatory response correlated with disease outcomes. Immune-modulatory agents, such as steroids, were subsequently employed as an adjunct to anti-viral agents to treat those hospitalized with SARS-CoV-2 [10,11]. Further understanding of COVID-19 pathogenesis led to a multimodal treatment approach targeting both viral and host immune response elements which ultimately showed superiority [12,13]. As the pandemic continued, the high morbidity and mortality caused by the SARS-CoV-2 virus, as well as the emergence of new viral variants, necessitated the development of novel treatment approaches [14]. This instigated the development of COVID-19 specific monoclonal antibodies, with the goal of preventing progression of infection to severe disease and hospitalization [15]. Monoclonal antibodies recognizing various viral proteins, primarily spike proteins, neutralize the SARS-CoV-2 virus, thereby blocking viral entry into host cells [15]. Though the emergence of novel treatments improved outcomes for hospitalized patients and proved efficacious at preventing severe disease in non-hospitalized patients, the development of COVID-19 vaccines was a groundbreaking discovery that greatly altered the course of the pandemic [16]. This review will focus on the current treatment options for COVID-19 in the United States.

## 2. Clinical Trials of Therapeutics for COVID-19 Pneumonia

Since the onset of the COVID-19 pandemic, many therapies were proposed and trialed in efforts to minimize disease burden [17,18,19]. Current therapies focus on targeting the SARS-CoV-2 virus (Table 1) or on modulating the host inflammatory response to the virus (Table 2). Though many therapeutics have been trialed, data are inconsistent as to the efficacy of many of these therapies, especially as various viral strains have emerged. Here we review the major clinical trials that evaluated the efficacy of the currently used therapies for COVID-19 in adults in the United States and discuss the molecular mechanisms by which these therapies target SARS-CoV-2 infection.

### 2.1. Antiviral Therapies

#### 2.1.1. Treatments Targeted toward Viral Replication

##### Remdesivir

Remdesivir, an inhibitor of the viral RNA-dependent, RNA polymerase, was one of the first antiviral agents used for the treatment of COVID-19. Remdesivir functions by blocking viral genomic replication. It was first used for treatment of the Ebola virus in 2014 and was later confirmed to have antiviral activity against SARS and MERS coronaviruses [20].

Spinner et al. conducted one of the first large-scale Randomized Controlled Trials (RCTs) studying this drug in COVID-19. They enrolled 584 patients who were randomized to receive a ten-day course of remdesivir, a five-day course of remdesivir or standard of care. Remdesivir was administered as a 200 mg dose on day one, followed by 100 mg on days two through five, or days two through ten. The results showed similar clinical outcomes between a 10-day course of remdesivir and standard of care (*p* = 0.18). Those treated with five days of remdesivir, however, were found to have a statistically higher odds of improved clinical status on day 11 compared with those receiving standard of care (*p* = 0.02) [21]. Following this, Beigel et al. conducted the Adaptive COVID-19 Treatment Trial (ACTT-1) trial [9]. The ACTT-1 trial enrolled 1062 participants from 60 trial sites across the world. Patients were randomly assigned to receive remdesivir (200 mg on day one followed by 100 mg on days two through ten) or a placebo. Assessment of clinical status on day 29, using an eight-point ordinal scale, showed a significantly shorter time to recovery in the remdesivir group compared with the placebo group (rate ratio 1.29 [95% CI: 1.12 to 1.49]), *p* < 0.001) [9]. Additionally, this study found that those treated with remdesivir were more likely to have improvement in an eight-point ordinal scale of disease severity at day 15. Though Kaplan–Meier assessment of mortality at day 29 was lower in the remdesivir group, there was no statistically significant difference in mortality between the two groups (Hazard Ratio 0.73 [95% CI: 0.52 to 1.03]). The authors noted that the mortality and recovery benefit of remdesivir was most evident in patients receiving minimal oxygen therapy prior to treatment, though patients with high ordinal scales at baseline also had an improvement in time to recovery. The researchers of this study also proposed that treatment with remdesivir may have prevented progression to severe disease, given that the treatment group had a lower proportion of oxygen-naive patients develop a need for supplemental oxygen during their hospitalization and a lower proportion of patients required higher levels of respiratory support. In March 2020, Goldman et al. and Gilead Sciences also conducted a multicenter RCT (the SIMPLE trial), which included 397 patients from 55 centers around the world. Patients in this study were randomized to receive remdesivir for five days versus ten days, both groups receiving 200 mg on day one followed by 100 mg on subsequent days. This trial found that clinical status on a seven-point ordinal scale was not significantly different between the groups at day 14 (*p* = 0.14), though notably, no control group was included in this study [22]. These results were similar even after adjustment for disease severity at the time of remdesivir administration. Based on these preliminary results, the US Food and Drug Administration (FDA) issued Emergency Use Authorization (EUA) of remdesivir on 1 May 2020 [9]. In May 2021, the WHO conducted the SOLIDARITY trial, which included 11,330 adults from 405 hospitals in 30 countries. Those assigned to remdesivir received the drug for ten days. Interim analysis showed that death occurred in 301 of 2743 patients receiving remdesivir and in 303 of 2708 receiving its control (*p* = 0.50) [23]. Recently, Gottlieb et al. conducted a double-blind placebo-controlled trial of non-hospitalization patients with COVID-19, aiming to assess the benefit of remdesivir at preventing disease progression and the need for hospitalization [24]. This study randomized 562 outpatients with COVID-19 to receive remdesivir (200 mg on day 1 and 100 mg on days 2 and 3) versus a placebo. These researchers found that among non-hospitalized patients with COVID-19, a 3-day course of remdesivir led to an 87% lower risk of hospitalization or death compared with the placebo.

In summary, despite largely promising results from initial large-scale studies showing an improvement in clinical status and prevention of progression to severe disease, no studies to date show a significant reduction in mortality with the use of remdesivir. Additionally, though previous studies suggested that remdesivir may provide greater benefit in those with less severe disease, these observations remain inconclusive and varied [21,25].

##### Paxlovid

More recently, oral antiviral agents have gained FDA EUA for use in COVID-19 [26,27]. Paxlovid, a combination of nirmatrelvir and ritonavir, was the first antiviral pill to gain EUA in the United States on 22 December 2021 [28]. Nirmatrelvir is a protease inhibitor of the SARS-CoV-2 M^pro^ enzyme that functions to prevent viral replication. Nirmatrelvir is metabolized by the CYP3A4 system; therefore, coadministration of nirmatrelvir with ritonavir, a CYP3A4 inhibitor, improves the pharmacokinetics of nirmatrelvir. Hammond et al. conducted the phase 2–3 RCT of Paxlovid, known as the EPIC-HR trial, whereby 2246 unvaccinated, non-hospitalized, COVID-19-positive patients were assigned to receive Paxlovid versus a placebo every 12 h for five days. At 28 days, the incidence of COVID-19-related death or hospitalizations was significantly lower in those who received Paxlovid compared with the placebo group by 6.32 percent (*p* < 0.001) [29].

##### Molnupiravir

Another recent oral antiviral medication, molnupiravir, also gained FDA EUA in December 2021. Molnupiravir is a small molecule ribonucleoside prodrug that functions by incorporating into viral DNA during viral replication and causing deleterious mutation in the viral RNA [30,31]. FDA EUA for this drug was based on results from the phase 3 clinical trial, MOVe-OUT. This study enrolled 1433 unvaccinated, non-hospitalized, COVID-19-positive participants and assigned them to receive 800 mg of molnupiravir versus placebo, twice daily for five days. Results of this study showed a 6.8% lower risk of hospitalization from any cause and death at day 29 (*p* = 0.001) in the interim analysis [32]. Though large-scale RCTs studying the efficacy of paxlovid and molnupiravir remain limited, results of these initial trials are promising. Effective anti-viral agents are a powerful treatment tool, especially when global vaccination is challenging to achieve. The availability of these oral antivirals provided a valuable treatment option to prevent disease progression in outpatients with a high risk for severe disease from COVID-19. However, obstacles for drug development, including the lack of suitable pre-clinical models for drug screening and testing, narrow treatment windows, and the constant mutation of SARS-CoV-2 remain and may hinder drug efficacy [33].

#### 2.1.2. Direct SARS-CoV-2-Neutralizing Monoclonal Antibodies

Early development of neutralizing monoclonal antibodies to SARS-CoV-2 viruses showed promising efficacy among non-critically patients [34,35,36]. These monoclonal antibodies specifically recognize the SARS-CoV-2 spike protein receptor domains that bind to ACE2, thereby inhibiting viral entry into host cells. However, their benefits are challenged by mutations on spike protein domains in new emerging variants such as delta, omicron, etc. New strategies of monoclonal antibody development for sustained efficacy in mutant variants are in a pre-clinical study phase [37,38]. The currently available SARS-CoV-2-neutralizing monoclonal antibodies available for clinical use are discussed as follows.

##### Bamlanivimab and Etesevimab

Bamlanivimab/etesevimab earned early EUA from the FDA based on the BLAZE-1 trials in November 2020 [39]. The first part of the BLAZE-1 trial was a phase 2–3 RCT conducted at 49 centers around the United States which studied non-hospitalized patients with mild to moderate disease severity who were at high risk for disease progression. Treatment groups were bamlanivimab, a combination of bamlanivimab and etesevimab, or a placebo. The combination monoclonal antibody group experienced a significant reduction in viral load at day 11 compared with the placebo. Conversely, no significant difference was seen in viral-load reduction in the bamlanivimab monotherapy arm at any dose compared to the placebo [39]. In the phase 3 BLAZE-1 trial, similar studied subjects were randomized to receive either a single intravenous infusion of bamlanivimab/etesevimab together or a placebo. At day 29, the monoclonal antibody group had a significant reduction in COVID-19-related hospitalization or death from any cause (*p* < 0.001) [40]. Though bamlanivimab was initially granted EUA both for use individually and in combination with etesevimab, the FDA changed its EUA in April 2021, approving use of bamlanivimab only in combination with etesevimab, given concerns that bamlanivimab alone was not effective against SARS-CoV-2 variants. Additionally, the most recent studies suggest that the combination of bamlanivimab and etesevimab may not be effective against the omicron variant, and as of January 2022, the National Institutes of Health (NIH) COVID-19 treatment guidelines panel recommended against the use of bamlanivimab/etesevimab, given reduced activity against the omicron variant [41].

##### Caririvimab and Imdevimab

Also in November 2020, the FDA granted EUA for casirivimab and imdevimab (REGEN-COV) administered together. EUA was based on a phase 1–3 trial by Weinrich et al. The phase 3 trial enrolled 4567 non-hospitalized participants with COVID-19, who were deemed at high risk of progression to severe disease; they were randomized to receive a single dose of 1200 mg REGEN-COV, 2400 mg REGEN-COV, or a placebo. Both the 1200 mg and 2400 mg REGEN-COV groups had a significant relative risk reduction in COVID-19 related hospitalization or death (*p* < 0.001 and *p* = 0.002, respectively). Moreover, patients in both treatment groups had a significantly lower median time to symptom resolution compared with the placebo [42]. REGEN-COV was also studied for use as post-exposure prophylaxis in a phase 3 trial by O’Brien et al. [43]. This study randomized 1505 participants who had a household contact positive for COVID-19 within the preceding 96 h, but without evidence of prior or ongoing COVID-19 infection themselves, to receive REGEN-COV versus a placebo. This study found that those who received REGEN-COV had a significant absolute risk reduction in COVID-19 symptom development [43]. The FDA initially granted EUA for the use of REGEN-COV in those with mild to moderate SARS-CoV-2 and high risk of progression to severe disease and for use as post-exposure prophylaxis for those with high risk of developing severe disease; nevertheless, the NIH COVID-19 treatment guidelines panel currently recommends against the use of REGEN-COV, given a lack of efficacy against the prominent omicron variant.

##### Sotrovimab

In May 2021, the FDA granted EUA for sotrovimab based on the results of the COMET-ICE trial [44]. This phase 3 trial randomly assigned 983 non-hospitalized patients with symptomatic COVID-19 infection to receive a single dose of sotrovimab (500 mg) versus a placebo. At 29 days, those who received sotrovimab had a significant relative risk reduction in hospitalization and death (*p* = 0.002) [44]. Notably, sotrovimab is also active against other coronaviruses and over the course of the COVID-19 pandemic, sotrovimab retained activity against variants of interest in in-vitro studies. This preservation of activity is possibly due to the binding of sotrovimab to a highly conserved epitope on the viral spike protein that is retained as the virus evolves and mutates [44]. This property is unique compared with other monoclonal antibodies. Additionally, sotrovimab contains two amino acid modifications in the Fc region of the antibody that are known to increase the half-life of the antibody, and which may improve bioavailability in the respiratory mucosa [44].

##### Bebtelovimab

Most recently, in February 2022, the FDA issued EUA for bebtelovimab. Like early monoclonal antibodies, bebtelovimab binds to the SARS-CoV-2 spike protein attachment site with the ACE2 receptor [45]. EUA was based on the results of the BLAZE-4 trial, a phase 1–2 trial assessing the use of bebtelovimab in unvaccinated outpatients with COVID-19. Notably, this study was performed before the emergence of the omicron variant. Though the results of this study are currently unpublished, they are referenced in the bebtelovimab FDA EUA fact sheet. The BLAZE-4 trial contained multiple treatment arms, thereby aiming to assess the efficacy of bebtelovimab in individuals with both a low and high risk of progression to severe disease. Participants in the phase 2 placebo-controlled subset were deemed at low risk of disease progression. Participants were randomized to receive a single infusion of 700 mg bamlanivimab, 1400 mg etesevimab, and 175 mg bebtelovimab (three-antibody regimen) versus 75 mg bebtelovimab alone versus a placebo. There was no significant difference seen in viral load at day 7 or hospitalization due to COVID-19, or death at day 29, between the groups. Those in the bebtelovimab group did have a significantly shorter time to symptom recovery compared with placebo (*p* = 0.003), whereas those with the three-antibody regimen had no significant difference in time to symptom recovery compared to placebo (*p* = 0.289). The authors suspected that no significant difference in hospitalization or death was seen in this population, given their already low risk of disease progression. Though a subset of this study did assess individuals at high risk of severe disease, this subset randomized participants to receive bebtelovimab versus the three-antibody regimen and no placebo and was included in this aspect of the study. Based on this initial data, bebtelovimab is currently granted EUA for use in individuals with mild to moderate COVID-19 and who are deemed at high risk for progression to severe disease [45]. Unlike many other COVID-19 therapies which have been studied in hospitalized individuals with moderate to severe COVID-19, the development of SARS-CoV-2 specific monoclonal antibodies provides treatment options for outpatients with the goal of preventing severe disease.

To this end, the strategy to decelerate viral replication or eradicate viruses seems to be an effective approach, considering that global vaccination cannot be easily achieved. However, identifying an effective agent and a proper timing of administration remains challenging. Moreover, the constant mutation of coronaviruses creates an additional impediment for this approach to effectively curb the COVID-19 pandemic.

### 2.2. Immunomodulatory Therapies

#### 2.2.1. Glucocorticoids

The basic pathogenesis of SARS-CoV-2 infection is excessive host immune responses, a phenomenon described as a “cytokine storm” that is associated with early and rapid multi-system organ damage, particularly among vulnerable individuals [46]. The hyperinflammatory response, primarily caused by type I interferon dysregulation, mediates macrophage and monocyte-derived macrophage activation, triggering multiple downstream inflammation regulatory pathways, thereby causing the massive release of multiple interleukin cytokines, including interleukine-6 (IL-6) [47]. Additionally, lymphopenia, a hallmark feature of COVID-19, was shown to directly correlate with disease severity [48]. As described by Yang et al., SARS-CoV-2 may cause lymphocyte depletion by directly infecting T cells via the ACE2 receptor [49].

The anti-inflammatory property of glucocorticoids is the basis for its use in COVID-19, functioning to hinder the hyper-inflammatory response state. Glucocorticoids were previously investigated for use in SARS and MERS; however, results from small, largely observational studies, were inconclusive and varied [50]. The most widely used glucocorticoid for treatment of COVID-19 in the United States is dexamethasone. The RECOVERY trial was the first large-scale RCT to investigate the use of dexamethasone in COVID-19 [51]. Among the 6425 patients that were enrolled in this study, 2104 were randomized to receive dexamethasone (6 mg once daily for 10 days) and 4321 received standard of care. The all-cause mortality was lower in the dexamethasone group compared with the standard of care group at 28 days (rate ratio, 0.83 [95% CI: 0.75 to 0.93], *p* < 0.001). This reduction in death rate was not dependent on mechanical ventilator support but dexamethasone showed no significant mortality benefit in patients not requiring any respiratory support [51]. Similarly, the observational cohort study by Crothers et al. demonstrated no improvement in mortality in those receiving low-flow or no supplemental oxygen [52]. The CoDEX trial showed a beneficial role of dexamethasone, specifically in the critically ill population [53]. This multicenter RCT was conducted in 41 intensive care units (ICUs) in Brazil. Patients were randomized to receive 20 mg of dexamethasone daily for five days, followed by 10 mg dexamethasone daily for five days, or until ICU discharge versus standard of care. At 28 days, those in the dexamethasone group had more ventilator-free days (*p* = 0.04); however, no significant benefit was seen in all-cause mortality or ICU-free days [53]. Based on these trials, the NIH currently recommends use of dexamethasone only in hospitalized patients with COVID-19 requiring supplemental oxygen.


ijms-23-07702-t001_Table 1Table 1Studies Evaluating the Effect of the Currently Used Anti-Viral Therapies Against COVID-19 in the United States.
Study

Year

Study Design

Setting

N

Treatment

Days to Primary Outcome

Primary Outcome(s)

Findings


**Drugs Targeting Viral Replication**

Remdesivir Spinner et al. [21]2020* RCT* IP 58410-day vs. 5-day of remdesivir vs. SOC10 days Clinical Status Improved clinical status in the 5-day remdesivir group (*p* = 0.02)ACTT-1 [9]2020RCT IP106210-days Remdesivir vs. placebo 28 daysClinical status Improved clinical status (*p* < 0.001)SIMPLE Trial [22]2020RCT IP39710-day vs. 5-day of remdesivir 14 days Clinical statusNo difference between groups (*p* = 0.14)SOLIDARITY [23]2021RCT IP11,33010-day remdesivir vs. no trial drug28 days Mortality No difference between groups (*p* = 0.50)
**Paxlovid**
EPIC-HR [29]2021RCT * OP2246Paxlovid vs. placebo for 5 days28 days COVID related hospitalization or deathLower primary outcome in treatment group (*p* < 0.001)
**Molnupiravir **
MOVe-OUT [32]2021 RCT OP1433Molnupiravir vs. placebo for 5 days 28 days COVID related hospitalization and death Lower primary outcome in treatment group (*p* = 0.001)

**Monoclonal Antibodies**


**Bamlanivimab and Etesevimab **
BLAZE-1 Phase 3 Trial [40]2021RCT OP1035Bamlanivimab + etesevimab vs. placebo single infusion 28 days COVID related hospitalization or deathReduction in primary outcome in treatment group (*p* < 0.001) 
**REGEN-CoV**
Weinrich et al. Phase 3 Trial [42]2021RCT OP4567 1200 mg vs. 2400 mg REGEN-CoV vs. placebo single dose29 days COVID related hospitalization or deathReduction in primary outcome in 1200 mg and 2400 mg treatment groups (*p* < 0.001 and *p* = 0.002 respectively) 
**Sotrovimab**
COMET-ICE [44]2021RCT OP983Sotrovimab vs. placebo single dose29 days COVID related hospitalization or death Reduction in relative risk of COVID-19 related hospitalization or death (*p* = 0.002)
**Bebtelovimab**
BLAZE-4 phase 2 [45] 2021RCT OP380bamlanivimab, etesevimab and (three-antibody regimen) versus bebtelovimab alone versus placebo as single dose7 days Persistently high viral load No significant difference between the groups* RCT = Randomized Controlled Trial; IP = In patient; OP = outpatient.


The COVID-STEROID 2 trial aimed to determine the efficacy of high versus low dose steroids in patients with severe hypoxemia, defined as requiring at least 10 L/min of supplemental oxygen [54]. This study randomized 1000 adults from 26 hospitals in Europe and India to receive 12 mg versus 6 mg dexamethasone for up to 10 days. At 28 days, there was no significant difference in mortality between the groups.

Smaller RCTs have also assessed the use of methylprednisolone for treatment of COVID-19 [55]. Corral-Gudino et al. conducted the GLUCOCOVID trial which found that the use of methylprednisolone was associated with a reduced risk of in-hospital death, admission to the ICU, or need for non-invasive ventilation (*p* = 0.024) [55].

Studies have also attempted to compare the efficacy of dexamethasone versus methylprednisolone. Pinzόn et al. randomized 216 patients to receive dexamethasone versus methylprednisolone (250 to 500 mg daily for three days, followed by prednisone 50 mg daily for 14 days) [56]. Results were notable for a higher percentage of patients developing ARDS, requiring transfer to the ICU, and a high percentage of mortality in the dexamethasone compared with the methylprednisolone group, suggesting a more superior effect of methylprednisolone [56]. A similar study was performed by Ranjbar et al. in Iran and, as in the previous study, these researchers found that those in the methylprednisolone group had a significantly better clinical status assessed by a 9-point WHO ordinal scale at both five and ten days (*p* = 0.002 and *p* = 0.001, respectively). There was also a significantly lower duration of hospital stay and a significantly lower progression to the need for mechanical ventilation in the methylprednisolone group [57]. Despite results from these trials suggesting a benefit of methylprednisolone over dexamethasone, routine use of methylprednisolone was not adopted in the United States, largely due to a lack of large RCTs. To date, glucocorticoids, primarily dexamethasone, remains the most effective, evidence-based agent to combat the hyperinflammatory response during COVID-19 infection.

#### 2.2.2. Janus Kinase Inhibitors

Janus kinase (JAK) is a regulatory enzyme which belongs to the tyrosine kinases family. JAK mediates many cytokine receptors including IL-2, IL-6, IL-10, interferon γ, and GM-CSF [58]. Hence, blocking this kinase enzyme can cause broader inflammatory inhibition than do IL-6 inhibitors. Baricitinib, a JAK inhibitor, currently approved by the FDA for use in rheumatoid arthritis, was used in severe COVID-19 cases. Following the results of the ACTT-1 trial that showed an improvement in clinical severity with the treatment of remdesivir, the ACTT-2 trial aimed to further improve outcomes in COVID-19 by targeting the inflammatory response [59]. This trial enrolled 1033 patients from 67 trial sites in 8 countries. Patients were randomized to receive either remdesivir and baricitinib (4 mg once daily) or remdesivir and a placebo for 14 days or until hospital discharge. The dose of baricitinib was reduced to 2 mg, once daily, for those with a glomerular filtration rate less than 60 mL per minute. The results indicated that those treated with remdesivir and baricitinib had a shorter median time to recovery (rate ratio 1.16 [95% CI: 1.01 to 1.32], *p* = 0.03), and a significantly higher odds of clinical improvement on day 15. When stratified by disease severity, those requiring non-invasive ventilation or high-flow oxygen had the largest benefit in time to recovery with the addition of baricitinib. However, there was no significant mortality benefit due to an inadequate power calculation [59]. Based on the promising results of ACTT-2, on 19 November 2020, the FDA issued EUA for baricitinib in combination with remdesivir for those with COVID-19 requiring hospitalization. Subsequently, the COV-BARRIER trial was conducted, which assessed the efficacy of baricitinib alone in contrast to standard of care. This was a phase 3 RCT and included 1525 participants from 101 centers in 12 countries. Participants were randomized to receive baricitinib 4 mg once daily versus a placebo for up to 14 days. This study found that 27.8% of patients in the baricitinib group and 30.5% of patients in the control group experienced disease progression at 28 days (*p* = 0.18). Though there was no significant difference in disease progression between the groups, a significant reduction in mortality was seen in the baricitinib group at 28 days (10% in baricitinib group vs. 13% in placebo group; HR 0.57, *p* = 0.0018) [60]. Given this reduction in mortality and a similar safety profile noted between baricitinib and the control group, the FDA revised its EUA for use of baricitinib alone on 28 July 2021, no longer requiring use in combination with remdesivir.

#### 2.2.3. Interleukine-6 Receptor Inhibitors

Tocilizumab and sarilumab are monoclonal antibodies that recognize the IL-6 receptor, thereby preventing IL-6 mediated inflammatory signaling pathways and responses [61,62]. Currently, tocilizumab is approved in the United States for use in a variety of autoimmune and inflammatory conditions, including rheumatoid arthritis, giant cell arteritis, and cytokine-release syndrome. Its use in the treatment of COVID-19 was first evaluated in the REMAP-CAP trial, which randomized patients to receive tocilizumab, sarilumab or standard of care in critically ill cases who required organ support measures [63]. Patients included in this study were limited to those requiring organ support in the ICU. This study found that that those in the tocilizumab and sarilumab groups had an increased number of days free of organ support and better survival at 90 days compared with the placebo group [63]. Subsequently, the Evaluating Minority Patients with Actemra (EMPACTA) and COVACTA RCTs were conducted to further evaluate the efficacy of tocilizumab in broader clinical severities of COVID-19-infected patients [64]. The EMPACTA trial found that in non-critically ill hospitalized cases, tocilizumab reduced the likelihood of progression to requirement of mechanical ventilation or death but without an overall 90-day survival benefit [64]. Following this, the COVACTA trial, a phase 3 RCT, assessed clinical status at 28 days after infusion of tocilizumab versus standard of care and found no significant improvement in clinical domains or reduction in mortality rate between the trial groups [65]. The largest study to evaluate the efficacy of tocilizumab was the RECOVERY trial that enrolled 4116 patients and randomized them to receive tocilizumab versus standard of care. Tocilizumab was given as a 400–800 mg dose, depending on weight, and administered either once or twice, about 12 to 24 h after the initial dose, if the patient’s condition had not improved. Tocilizumab was shown to have a 28-day mortality benefit (*p* = 0.0028), as well as improved length of hospital stay at 28 days (*p* < 0.0001). Finally, among patients not already on mechanical ventilation at the onset of the trial, tocilizumab was associated with less mechanical ventilation use or risk of death during the course of the study compared with those in the control group (*p* < 0.0001) [66]. One possible reason for the differing results seen between these studies is the sample size. The RECOVERY trial is the largest study to date assessing the efficacy of tocilizumab in COVID-19. Additionally, the above trials all vary in their enrollment criteria, ranging from those requiring organ support to excluding those requiring mechanical ventilation. Moreover, the variation of ‘standard of care’ treatments in the control group was a confounding factor as standard of care treatment guidelines varied over the course of the pandemic. Fewer studies have assessed the efficacy of Sarilumab and, currently, the NIH COVID-19 treatment guidelines recommend use of Sarilumab only when tocilizumab is unavailable, based on the positive findings of the REMAP-CAP trial [32].

In summary, the approaches to minimize tissue and organ damage caused by severe SARS-CoV-2 infection by curtailing excessive inflammatory response showed clinical benefits. However, the morbidity in non-critically ill patients and morbidity in critically ill patients remain substantial despite such treatments. Earlier treatments in non-critically ill populations showed only modest effects on limiting disease progression, suggesting a delicate balance of medically regulated host-defense mechanisms during inflammation.


ijms-23-07702-t002_Table 2Table 2Studies Evaluating the Effect of Currently Used Treatments Targeting the Inflammatory Response Caused by the SARS-CoV-2 Virus.
Study

Year

Design Type

Setting

N

Treatment

Duration to Primary Outcome

Primary

Outcome(s)

Findings


**Glucocorticoids**

RECOVERY [51]2020* RCT * IP6425Dexamethasone 6 mg for 10 days vs. SOC28 days All cause mortality Lower death rate in dexamethasone group (*p* < 0.001)CoDEX [53]2020RCTIP299Dexamethasone 20 mg for 5 days then 6 mg for 5 days vs. SOC28 days Ventilator free days More ventilator free days in dexamethasone group (*p* = 0.04)GLUCOCOVID [55]2020RCT IP85* MP 40 mg twice daily for 3 days then 20 mg twice daily for 3 days vs. SOCDuration of hospitalization Death, ICU admission or need for non-invasive ventilation Reduction in primary endpoint in treatment group (*p* = 0.024)Pinzόn et al. [56]2020Cohort IP216Dexamethasone vs. MP followed by dexamethasoneDuration of hospitalizationRecovery time Shorter recovery time in MP group (*p* < 0.0001)Ranjbar et al. [57]2020RCTIP86Dexamethasone vs. MP28 days all cause mortality. Clinical status on days 5 and 10All cause mortality and clinical statusImproved clinical status at days 5 and 10 in MP group (*p* = 0.002 and *p* = 0.001) respectively. No difference in mortality 

**JAK Inhibitors**

ACTT-2 [59]2020RCTIP 1033Remdesivir + placebo vs. remdesivir + baricitinib28 days Recovery timeRemdesivir + baricitinib had a shorter time to recovery (*p* = 0.03)COV-BARRIER [60] 2020RCT IP1525Baricitinib vs. SOC28 days Progression of disease or deathReduction in death in baricitinib group (*p* = 0.0018) but no difference in disease progression (*p* = 0.18) 

**IL-6 Receptor Inhibitors **

REMAP-CAP [63]2020RCTIP803Tocilizumab vs. sarilumab vs. SOC21 days Days free of organ supportIncreased days free of organ support in tocilizumab and sarilumab groups (posterior probabilities of superiority of more than 99.9% and of 99.5%, respectively)EMPACTA [64]2020RCTIP 389Tocilizumab vs. placebo28 days Mechanical ventilation or deathReduction in need for mechanical ventilation or death in tocilizumab group (*p* = 0.04)COVACTA [65]2020RCT IP452Tocilizumab vs. SOC 28 days Clinical status No significant improvement in clinical status between groups (*p* = 0.31)RECOVERY [66]2020–2021RCT IP 4116Tocilizumab vs. SOC 28 days All cause mortalityLower rate of death in the tocilizumab group *(p* = 0.0028)* RCT = Randomized controlled trial; SOC = Standard of Care; MP = Methylprednisolone; IP = Inpatient.


## 3. Conclusions

The swift global spread and high mortality among vulnerable populations caused by the COVID-19 pandemic necessitated the rapid development of preventative strategies and therapeutics against this novel virus. The focus of therapeutics research is largely twofold, aiming to target, either the SARS-CoV-2 virus directly, or the human inflammatory response that causes the cytokine storm and severe disease. Numerous therapeutics were investigated since the start of the COVID-19 pandemic and the emergence of new data from clinical trials has caused shifts in clinical practice. Currently, a combination of steroids and anti-viral medications, such as remdesivir, serve as the cornerstone of treatment for hospitalized patients with COVID-19 in the United States (Figure 1). For non-hospitalized patients, the more recent discovery of oral antiviral medications and newly developed monoclonal antibodies directed against SARS-CoV-2 have proven to be an efficacious treatment during an early stage of infection.

Most recently, the development of therapeutics focused on pre-exposure prophylaxis, especially as new strains of SARS-CoV-2 cause concern over vaccine efficacy toward these newer strains. The combination of tixagevimab plus cilgavimab (Evusheld) monoclonal antibody is the only therapeutic with EUA for pre-exposure prophylaxis and is limited for use in individuals who have not been exposed to COVID-19 but are immunocompromised or who cannot receive COVID-19 vaccination [67]. As new variants continue to emerge, there exists a need for continued research into additional monoclonal antibodies for outpatient COVID-19 treatment and prevention. Specifically, research focuses on creating antibodies against conserved epitopes on the SARS-CoV-2 virus. Moreover, studies involving cocktail combinations of monoclonal antibodies were proposed, aiming to target multiple sites on the SARS-CoV-2 virus [68]. Additionally, further studies are needed to identify biomarkers for treatment initiation and monitoring for an optimal dose and timing of treatment. Though many therapeutics were granted EUA by the FDA, no treatment to date serves as cure, largely due to the constant mutation of SARS-CoV-2 viruses. Vaccines against SARS-CoV-2 are our best defense against continued morbidity and mortality, routinely showing in clinical studies that vaccination reduces risk of hospitalization and severe disease. Despite this, many individuals remain unvaccinated, worldwide. Therefore, continued efforts are needed by healthcare providers to reinforce preventive measures and encourage mass vaccination.

In addition to the development of novel therapeutics, research focuses on identifying and minimizing the long-term complications of COVID-19 infection, including post-intensive care syndrome and recovery of chronic organ failure. Currently, the CDC and NIH have partnered to create the RECOVER initiative, a nation-wide study to learn about the long-term effects of COVID-19 in the United States.

A wealth of knowledge was generated over the course of the COVID-19 pandemic. Currently, large databases exist, allowing both healthcare providers and the public to access the most up-to-date information regarding COVID-19 treatments and drug trials. One such database is the COVID-19 Drug Repository, which includes clinical trial data, as well as drug specific molecular data for therapeutics developed around the world [69]. This database contains about 460 drugs, 184 which are approved for use and an additional 384 which are currently being investigated. By providing both approved and investigational drugs, this database aims to link drugs to PubMed and related research sources for easier information mining. The DockCoV2 drug database, created by Chen et al. and approved by both the FDA and the Taiwan National Health Insurance (NHI), aims to speed up drug discovery by predicting the binding affinity of drugs to seven key viral proteins known to function in viral replication and spread [70]. This database also provides information about drug activity against the SARS and MERS viruses. Currently, this database contains 3109 drugs. Lastly, the COVIDrugNet provides a database of ongoing drug research, intending to aid researchers in finding relevant drug trial studies [71]. Furthermore, this database provides information on drug structure and molecular target(s). In addition to these large databases, online drug literature and data platforms, such as LitCovid hub and DrugBank, are currently in existence, also allowing for easier data gathering [71,72]. As the COVID-19 pandemic continues, large databases such as these are a valuable asset, facilitating information sharing and global collaboration, with the goal of using existing knowledge to create newer therapeutic and preventive strategies against the SARS-CoV-2 virus.

## Figures and Tables

**Figure 1 ijms-23-07702-f001:**
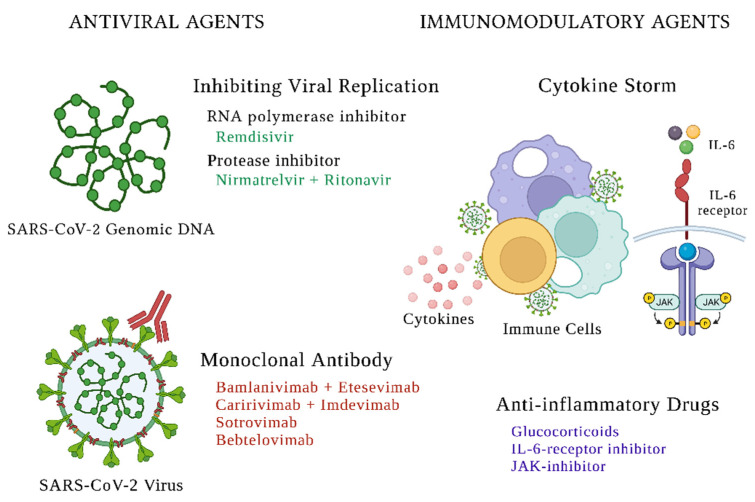
Mechanism of Action of Therapies Targeted Toward the SARS-CoV-2 Virus.

## Data Availability

No new data were created or analyzed in this study. Data sharing is not applicable to this article.

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
