# Peer review of "Current Therapeutics for COVID-19, What We Know about the Molecular Mechanism and Efficacy of Treatments for This Novel Virus"

_ijms, 2022, doi:10.3390/ijms23147702_

Round 1

Reviewer 1 Report

The article ‘Current Therapeutics for COVID-19, What We Know About the Molecular Mechanism and Efficacy of Treatments for This Novel Virus’ by Narayanan and Parimon summarizes the current treatment options available for COVID-19. The review article is structured nicely, and the work is relevant for the readership of ‘IJMS.’ However, the authors need to address some critical concerns in the current version of the manuscript before its publication. In addition, the figure and figure legend are superficial. The major and minor issues are listed below.

Major:

In the introduction section, the rationale behind using multiple therapeutic approaches such as inhibitors, monoclonal antibodies, or antiviral therapies is underdeveloped. Please lay out the rationale highlighting different mechanisms of action against the SARS-Co-V2 briefly before discussing the treatments. 

Figure 1 is not very informative and is high resolution. Can the authors confirm that the figure was created in a version of BioRender that supports publication-quality images? 

The authors have not discussed the efficacy of these drugs against different disease severity/immunotype s(1, 2, 3) or comorbid conditions. Information regarding these studies would be helpful to include when gauging the overall efficacy of drugs or designing new therapies for future variants. 

The authors have discussed all other therapeutic approaches; however, they have missed a discussion on vaccines altogether. A list of FDA EUA vaccines and their modes of action would be helpful to include. 

While the authors have attempted to summarize the relevant treatment strategies available in the field, they have not discussed any new hypotheses, open questions, or strategies to tackle them for future treatment development. 

Minor:

Please replace SARS-CoV2 with ‘SARS-Co-V2’. 

Line 122: Did the authors mean versus here?

Several other typographical/copy editing mistakes present throughout the paper need to be corrected in the revised version of the manuscript.

Author Response

Dear Editors,

We want to thank you for the opportunity to revise a manuscript to be considered for submission. We thank the reviewers for their valuable and constructive comments that helped tremendously to improve the manuscript. We provided our responses point-by-point below.

Reviewer # 1:

Major:

Point 1: In the introduction section, the rationale behind using multiple therapeutic approaches such as inhibitors, monoclonal antibodies, or antiviral therapies is underdeveloped. Please lay out the rationale highlighting different mechanisms of action against the SARS-Co-V-2 briefly before discussing the treatments. 

Response 1: The final paragraph in the introduction has been edited to include the rational for using multiple treatment modalities and briefly discussing their mechanisms of action.

Pont 2: Figure 1 is not very informative and is high resolution. Can the authors confirm that the figure was created in a version of BioRender that supports publication-quality images? 

Response 2: Figure 1 has been replaced in publication quality. Publication license from biorender is available

Point 3: The authors have not discussed the efficacy of these drugs against different disease severity/immunotype s(1, 2, 3) or comorbid conditions. Information regarding these studies would be helpful to include when gauging the overall efficacy of drugs or designing new therapies for future variants. 

Response 3: Additional information has been added discussing the studies that commented on drug efficacy based on disease severity.

Point 4: The authors have discussed all other therapeutic approaches; however, they have missed a discussion on vaccines altogether. A list of FDA EUA vaccines and their modes of action would be helpful to include. 

Response 4: In this review, we aimed to discuss treatment options for those already infected with SARS-CoV-2. Though vaccines are briefly mentioned as a prevention strategy that greatly altered the course of the pandemic, their mechanisms and trials regarding their efficacy were purposely not discussed in detail given that they fall under preventive measures rather than treatments for the virus.

Point 5: While the authors have attempted to summarize the relevant treatment strategies available in the field, they have not discussed any new hypotheses, open questions, or strategies to tackle them for future treatment development. 

Response 5: More discussion on future directions and research aims have been added to the conclusion section 

Minor:

Point 6: Please replace SARS-CoV2 with ‘SARS-Co-V2’

Response 6: SARS-CoV2 has been replaced with SARS-CoV-2 as is typically written in the nomenclature

Point 7: Line 122: Did the authors mean versus here?

Response 7: The word is in line 140 (due to edits) has been changed from verses to versus.

Point 8: Several other typographical/copy editing mistakes present throughout the paper need to be corrected in the revised version of the manuscript.

Response 8: Identified typos have been corrected.

Reviewer 2 Report

The authors performed a comprehensive review of current therapeutics for COVID-19 and described their molecular mechanism and efficacy. Overall the manuscript is informative. 

Major comments

(1) Although the authors described each section comprehensively, they didn't cite several recent relevant papers. Only 36 references are in this review. I would suggest adding more references as support.

(2) Recently, a few databases related to the COVID-19 ongoing treatments and drugs have been developed and published. Such as "COVID-19 Drug Repository", "COVIDrugNet" and "DockCoV2".

(i) Tworowski, Dmitry, et al. "COVID19 Drug Repository: text-mining the literature in search of putative COVID19 therapeutics." Nucleic acids research 49.D1 (2021): D1113-D1121.

(ii) Menestrina, Luca, Chiara Cabrelle, and Maurizio Recanatini. "COVIDrugNet: a network-based web tool to investigate the drugs currently in clinical trial to contrast COVID-19." Scientific reports 11.1 (2021): 1-15.

(iii) Chen, Ting-Fu, et al. "DockCoV2: a drug database against SARS-CoV-2." Nucleic Acids Research 49.D1 (2021): D1152-D1159.

Authors should describe these databases, which could help researchers to find more relevant information on current therapeutics for COVID-19.

Author Response

Dear Editors,

We want to thank you for the opportunity to revise a manuscript to be considered for submission. We thank the reviewers for their valuable and constructive comments that helped tremendously to improve the manuscript. We provided our responses point-by-point below.

Reviewer # 2:

Major:

Comment 1: Although the authors described each section comprehensively, they didn't cite several recent relevant papers. Only 36 references are in this review. I would suggest adding more references as support.

Response 1: Additional references have been added

Comment 2: Recently, a few databases related to the COVID-19 ongoing treatments and drugs have been developed and published. Such as "COVID-19 Drug Repository", "COVIDrugNet" and "DockCoV2".

(i) Tworowski, Dmitry, et al. "COVID19 Drug Repository: text-mining the literature in search of putative COVID19 therapeutics." Nucleic acids research 49.D1 (2021): D1113-D1121.

(ii) Menestrina, Luca, Chiara Cabrelle, and Maurizio Recanatini. "COVIDrugNet: a network-based web tool to investigate the drugs currently in clinical trial to contrast COVID-19." Scientific reports 11.1 (2021): 1-15.

(iii) Chen, Ting-Fu, et al. "DockCoV2: a drug database against SARS-CoV-2." Nucleic Acids Research 49.D1 (2021): D1152-D1159.

Authors should describe these databases, which could help researchers to find more relevant information on current therapeutics for COVID-19.

Response 2: Discussion of the above-mentioned databases as well as online platforms has been added to the conclusions.

Round 2

Reviewer 1 Report

After careful examination of the revised manuscript, the response of the authors to previous reviews, and the changes made in the manuscript, I gather that the revised version of the manuscript has addressed the major concerns raised in the previous version of the paper (the limitations about unresolved comments are understandable). Hence, I endorse the publication of this paper. 

Reviewer 2 Report

The authors substantially improved the manuscript.